

# A block cipher algorithm identification scheme based on hybrid k-nearest neighbor and random forest algorithm

Ke Yuan[1,2], Daoming Yu[1], Jingkai Feng[3], Longwei Yang[1], Chunfu Jia[4] and Yiwang Huang[5]

[1] School of Computer and Information Engineering, Henan University, Kaifeng, Henan, China
[2] Henan Key Laboratory of Big Data Analysis and Processing, Henan University, Kaifeng, Henan, China
[3] International Education College, Henan University, Zhengzhou, Henan, China
[4] College of Cybersecurity, Nankai University, Tianjin, Tianjin, China
[5] School of Data Science, Tongren University, Tongren, Guizhou, China

## ABSTRACT

Cryptographic algorithm identification, which refers to analyzing and identifying the encryption algorithm used in cryptographic system, is of great significance to cryptanalysis. In order to improve the accuracy of identification work, this article proposes a new ensemble learning-based model named hybrid k-nearest neighbor and random forest (HKNNRF), and constructs a block cipher algorithm identification scheme. In the ciphertext-only scenario, we use NIST randomness test methods to extract ciphertext features, and carry out binary-classification and five-classification experiments on the block cipher algorithms using proposed scheme. Experiments show that when the ciphertext size and other experimental conditions are the same, compared with the baselines, the HKNNRF model has higher classification accuracy. Specifically, the average binary-classification identification accuracy of HKNNRF is 69.5%, which is 13%, 12.5%, and 10% higher than the single-layer support vector machine (SVM), k-nearest neighbor (KNN), and random forest (RF) respectively. The five-classification identification accuracy can reach 34%, which is higher than the 21% accuracy of KNN, the 22% accuracy of RF and the 23% accuracy of SVM respectively under the same experimental conditions.

## INTRODUCTION

With the widespread application of cryptographic algorithms, their security has attracted a lot of people' attention. Most of the current cryptanalysis techniques are based on Kerckhoffs' principle, that is, under the premise that we have known the exact cryptographic algorithms used in the system first, then the analysis of various cryptographic algorithms is carried out. However, in practical applications, researchers and cryptanalysts usually can only obtain ciphertexts without knowing the exact cryptographic algorithms, and as time goes by, more and more cryptographic algorithms have emerged. In this situation, researchers can only perform ciphertext-only analysis. Therefore, identifying the cryptographic algorithm that is used by the ciphertext in

Corresponding author
Longwei Yang, ylw@henu.edu.cn

ciphertext-only scenario has become a prerequisite for cryptanalysis. Moreover, the research on cryptographic algorithm identification has very important theoretical significance, practical value, and the anti-identification ability can also be used as an evaluation of the security performance of one cryptographic algorithm correspondingly.

The existing cryptographic algorithm identification schemes have the following three characteristics:

1. The research object is mainly the block cipher algorithms.
2. Existing schemes mainly identify the block cipher algorithms that work in Electronic Code book (ECB) or Cipher Block Chaining (CBC) modes, and we can find the ciphertexts obtained in ECB mode are easier to identify than in CBC mode. *Zhao, Zhao & Liu (2018)* used RF model to identify the ciphertexts generated in ECB working mode.
3. Most of the current cryptographic algorithm identification schemes are based on a single-layer classifier model. *Mello & Xexéo (2016)* used the single-layer SVM classifier model to identify cryptographic algorithms; *Fan & Zhao (2019)* used three classification models: RF, logistic regression (LR) and SVM to carry out identification tasks on eight block cipher algorithms. However, we find that the single-layer classifier model has problems such as low efficiency when dealing with classification tasks, overfitting when samples have excessive noise, high training cost, curse of dimensionality, and difficulty in finding suitable parameters and kernel functions.

In order to solve the existing problems of the single-layer classifier model, this article proposes a new model based on the idea of ensemble learning to identify block cipher algorithms. The specific contributions of this work are as follows:

1. The HKNNRF block cipher algorithm identification model is proposed. The ensemble learning-based model HKNNRF overcomes the problems of the single-layer classifier model. It has the advantages of stronger, higher identification efficiency, and more flexible parameter setting.
2. Feature extraction with randomness test. Select AES, 3DES, Blowfish, CAST and RC2, these five block cipher algorithms as our research objects, and we set five ciphertext sizes, including 1, 8, 64, 256, 512 KB. Then, in the ciphertext-only scenario, we used NIST randomness test methods to extract ciphertext features.
3. Construct a new identification scheme of block cipher algorithm based on HKNNRF model. In the ciphertext-only scenario, we used the NIST randomness test methods to extract ciphertext features, and used the HKNNRF model to build an identification classifier, and perform binary and five classification experiments on block cipher algorithms. The experiments show that the proposed model has better performance than traditional machine learning models.

The rest of the article is arranged as follows. The first section introduces the existing related work. The second section discusses the principles of cryptographic algorithm identification. The third section introduces the RF algorithm, KNN algorithm and HKNNRF classification model. The fourth section lists the relevant data preparation work

and the evaluation criteria for classification results. The fifth section gives the experiment results and analysis. At last, we summarize the current work and put forward some prospects for the future work.

## RELATED WORK

In the early identification work on cryptographic algorithms, the main research object was classical cryptographic algorithms. *Pooja (2001)* designed a classical cryptographic algorithms identification scheme based on the frequency of letter usage. However, with the rapid development of modern cryptographic technology, more advanced symmetric cryptographic algorithms and asymmetric cryptographic algorithms have emerged, they are more complex than the intuitive classical mapping cryptographic algorithms. In this situation, traditional cryptographic algorithm identification scheme, which is based on statistical methods, is gradually failing. To cope with this problem and realize the efficient identification of cryptographic algorithms, researchers have used a series of machine learning algorithms to construct new identification schemes, such as SVM, KNN, and RF. The optimization of machine learning algorithms can use the Farmland fertility algorithm (*Gharehchopogh, Farnad & Alizadeh, 2021*), African Vultures Optimization Algorithm (*Abdollahzadeh, Gharehchopogh & Mirjalili, 2021a*), and Artificial Gorilla Troops Optimizer (*Abdollahzadeh, Gharehchopogh & Mirjalili, 2021b*). *Dileep & Sekhar (2006)* proposed a method to identify block cipher algorithms using SVM, they identified the ciphertexts that are encrypted by AES, DES, 3DES, RC5 and Blowfish in ECB and CBC modes. Experiments show that the SVM model using Gaussian kernel function has the best performance, and they got better identification accuracy in ECB mode than in CBC mode. *Sharif, Kuncheva & Mansoor (2010)* compared the identification performance of different classification models on block cipher algorithms, and they believed that the Rotation Forest model has better performance. *Chou, Lin & Cheng (2012)* used SVM model in their work to identify the ciphertexts that are encrypted by AES and DES in ECB and CBC modes. *Mishra & Bhattacharjya (2013)* proposed a block cipher and stream cryptographic algorithm identification scheme combining pattern recognition and decision tree, and greatly improved the identification accuracy using methods such as block length detection and reoccurrence analysis. *Wu, Wang & Li (2015)* proposed a hierarchical identification scheme based on k-means clustering algorithm, and had an identification accuracy of about 90% for classic block cipher algorithms. *Ding & Tan (2016)* identified five block cipher algorithms in pairs. After analyzing the results, they found that the setting of the key has a certain impact on the identification accuracy. *Mello & Xexéo (2018)* analyzed the ciphertexts encrypted by seven cryptographic algorithms of ARC4, Blowfish, DES, Rijdael, RSA, Serpent, and Twofish in both ECB and CBC modes, and they found these algorithms almost all can be fully identified in ECB mode.

## THE PRINCIPLES OF CRYPTOGRAPHIC ALGORITHM IDENTIFICATION

At present, the design ideas of cryptographic algorithm identification schemes mainly come from statistical methods and machine learning techniques. *Wu et al. (2015)* believed

that there are certain differences in the ciphertext space generated by different cryptographic algorithms, and the characteristics that characterize the corresponding differences can be extracted from them as the basis for the cryptographic algorithm identification task. The task is to distinguish ciphertext features with small differences, so as to identify the cryptographic algorithm corresponding to the specific ciphertexts.

In the early work of cryptographic algorithm identification, most researchers used identification schemes based on statistical methods. They select some specific statistical indicators, and compare them to get the final identification result. These statistics-based schemes are more intuitive and easier to understand. However, such schemes are only suitable for the scenario that there are fewer cryptographic algorithms. As time goes by, cryptography continues to develop, more and more cryptographic algorithms are proposed, it is precisely because of the increase in cryptographic algorithms, there are more complex and subtle differences between various ciphertext features that the statistics-based schemes cannot handle well. To cope with this problem and realize the efficient identification of cryptographic algorithms, a new kind of identification scheme, based on machine learning, steps into (*Gharehchopogh, 2022*; *Ghafori & Gharehchopogh, 2021*). In this new identification scheme, we regard ciphertext features as a set of attributes, the identification task is equivalent to the classification task, and it is completed in two stages. The first stage trains the classifier model on the training data set with ciphertext features and algorithm labels, the second stage is to identify the test set with the trained classifier, then get the identification result and evaluate it. Compared with the statistics-based identification scheme, the scheme based on machine learning is simple in design and stable in performance, and the most important thing is that it can handle complex data relationships. It is precisely because of these advantages that researchers are willing to invest more time and effort into it, there is no doubt that machine learning has become a research hotspot in the cryptographic algorithm identification field.

However, in the long run, the machine learning-based schemes also have obvious shortcomings, that is, it simply puts the cryptographic algorithm identification problem within the framework of general pattern recognition, and cannot discuss the particularity of cryptographic algorithm identification in depth, which brings difficulties to technological innovation.

The following gives the formal definition of the basic elements of cryptographic algorithm identification: ciphertext, ciphertext features, cryptographic algorithm identification and cryptographic algorithm identification scheme.

**Definition 1 Ciphertext.** Set cryptographic algorithm set $A = \{a_1, a_2, ..., a_n\}$, where $n$ is the number of cryptographic algorithms. For any given cryptographic algorithm $a_i$, there is a ciphertext file $C = \{c_1, c_2, ..., c_i\}$ generated by encrypting plaintext in *mod* mode, where $c_i$ is the ciphertext fragment of ciphertext file $C$.

**Definition 2 Ciphertext features.** Extract the features of the ciphertext file of unknown cryptographic algorithm, and obtain the $d$ dimension eigenvector $v = \{x_1, x_2, ..., x_d\}$. The ciphertext feature extraction process is expressed as the process of mapping ciphertext file $C$ into eigenvector $v$, which is denoted as $Withdraw(C) \rightarrow v$. *Withdraw* is the specific feature extraction process or method, also known as processing function. In the
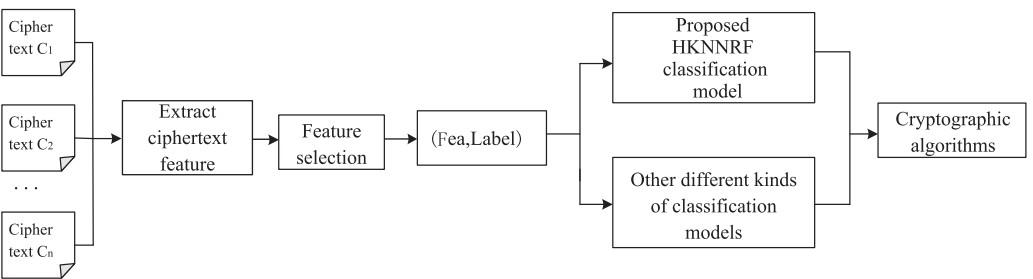

**Figure 1 Workflow of cryptographic algorithm identification.**

cryptographic algorithm identification scheme, the features of $C$ can be expressed as triple $fea = (E, Withdraw, d)$, where $E$ is the process of representing $C$ as a binary bit string.

**Definition 3 Cryptographic algorithm identification.** Known cryptographic algorithm set $A = (a_1, a_2, \ldots, a_n)$, ciphertext file $C$. Assuming that there is an identification scheme $P$, in the ciphertext-only scenario, identify the cryptographic algorithm $a_i$ adopted by $C$ with accuracy $h^p$, where $a_i \in A$. The whole process is called cryptographic algorithm identification, which is denoted as triple $R = (A, P, h^p)$. Its workflow chart is shown in Fig. 1.

**Definition 4 Cryptographic algorithm identification scheme.** In cryptographic algorithm identification $R = (A, P, h^p)$, assuming $oper$ is the process of $R$, $fea$ is the ciphertext feature extracted from ciphertext file $C$, $CLA$ is the classification algorithm used in the identification scheme, which refers to the subsequent SVM, KNN, RF and HKNNRF algorithms involved in this article, then the triple $P = (oper, fea, CLA)$ is denoted as a cryptographic algorithm identification scheme.

# A BLOCK CIPHER ALGORITHM IDENTIFICATION SCHEME BASED ON HKNNRF ALGORITHM

We propose the HKNNRF model for block cipher algorithm identification and selected five block cipher algorithms: AES, 3DES, Blowfish, Cast and RC2 as the research objects. In order to realize the effective identification of different cryptographic algorithms, our work was mainly divided into two stages. In the first stage, we used the specified method below to extract the features hidden in the ciphertext. In the second stage, we constructed the HKNNRF classifier proposed in this article. Then, we inputted the features into the classifier to complete the identification task. In this article, we selected 15 randomness test methods in the NIST randomness test package as the basic methods of ciphertext feature extraction, and selected 10 useful ciphertext features as the final features for classification.

## Random forest algorithm

The random forest (RF) model is an ensemble learning algorithm. Decision tree is the basic unit of random forest, RF constructs multiple decision trees (*Shen et al., 2018*) and integrates them to obtain the best decision results.

Selection measure of split attribute in decision tree

(1) Expectation of any classification sample of random forest model:

$$H(s_1, s_2, \ldots, s_m) = -\sum_{i=1}^{n} p_i \log_2(p_i).$$

The possible value of random variable $S$ could be one of $s_1, s_2, \ldots, s_m$, and for each possible value $s_i$, its probability

$$P(S = s_i) = p_i, (i = 1, 2, \ldots, m).$$

(2) The entropy of the sample set $S$ is fixed before division. Use the feature $A$ to segment $S$ and calculate the entropy of the segmented subset. The smaller entropy of $(s_1, s_2, \ldots, s_m)$, the smaller uncertainty of the subset that divided by feature $A$, and the better classification result.

(3) The sample set $S$ is divided into $I$ subsets $s_1, s_2, \ldots, s_i$ by features $A$.

$$H(S \mid A) = \sum_{i=1}^{n} \frac{|S_i|}{|S|} H(S_i) = -\sum_{i=1}^{n} \frac{|S_i|}{|S|} H(S_i) \sum_{k=1}^{k} \frac{|S_{ik}|}{|S_i|} \log_2 \left( \frac{|S_{ik}|}{|S_i|} \right).$$

(4) Information gain

$$Gain(S, A) = H(S) - H(S|A).$$

When input the sample set $S$ which are going to be classified, every decision tree is created in parallel and does not need pruning.

## K-nearest neighbor algorithm

In the k-nearest neighbor algorithm (KNN), all training samples are stored in Euclidean space as one point, and the training data are grouped, then we classify the samples to be predicted according to the category labels of the nearest K samples. The distance measurement of k-nearest neighbor algorithm generally adopts Euclidean distance, of course, it can also be other distances (*Li, 2018*).

Based on the Euclidean distance function $d(x_i, x_j)$, we can obtain the nearest $K$ adjacent nodes of this node, where the Euclidean distance function is:

$$d(x_i, x_j) = \sqrt{(x_{i,1} - x_{j,1})^2 + \ldots + (x_{i,m} - x_{j,m})^2}.$$

When new data without label appear, we can use the k-nearest neighbor algorithm to identify the new data according to the nearest sample labels in the known sample set. K-nearest neighbor algorithm is widely applied because it is simple, effective and easy to implement.

## Cryptographic algorithm identification classifier based on HKNNRF algorithm

Compared with the traditional classification model, although the RF greatly improves the classification accuracy, it is easily affected by the features that have been segmented many

01. Divide data source into train set and test set randomly, 80% as train set, 20% as test set. Then get (X_train, X_test, Y_train, Y_test)
02. Furthermore, get (X_train_RF, X_train_KNN, Y_train_RF, Y_train_KNN) by dividing (X_train, Y_train) into two equal parts randomly.
03. for tree_num in range tree_num_begin to tree_num_end:
     for tree_depth in range depth_begin to depth_end:
       for neighbor_num in range neighbor_num_begin to neighbor_num_end:
04.        Initialize RandomForest RF mainly by (tree_num,tree_depth)
05.        Initialize OneHotEncoder OHE
06.        Initialize KNN by neighbor_num
07.        Train RF with (X_train_RF,Y_train_RF)
08.        Fit OHE with returned leaf_indice_matrix after applying trees in RF to X_train_RF
09.        Get intermediate_feature_matrix after applying trees in RF to X_train_KNN
10.        Transform intermediate_feature_matrix by OHE to get New_Features
11.        Train KNN with (New_Features,Y_train_KNN) to get final HKNNRF
12.        Conduct binary and five classification experiments on (X_test,Y_test)
13.        Evaluate performance of current HKNNRF model
       end
     end
   end

**Figure 2 The cryptographic algorithm identification process based on HKNNRF.**

times. Based on this problem, this article attempts to propose an identification scheme of hybrid k-nearest neighbor and random forest algorithm based on the idea of stacking in ensemble learning (*Benyamin, Farhad & Saeid, 2021*; *Goldanloo & Gharehchopogh, 2022*). The idea of stacking refers to letting the output of the first layer classifier as the input of the next layer's classifier. By analogy, we can get the final classification result by stacking.

The pseudocode flowchart of the HKNNRF model is shown in Fig. 2. The algorithm first uses the original features to train the RF model, then uses the decision trees in RF to construct new features, add them to the original features to get final features, then normalize the final features with one-hot encoder. Finally, input the normalized features into KNN for training.

We selected the cryptographic algorithm identification scheme $P = (oper, fea, HKNNRF)$ defined in definition 4, Its workflow is as follows.

Input: A set of cryptographic algorithms $A = \{a_1, a_2, \ldots a_k\}$ ( $k$ is the number of cryptographic algorithms), the number of ciphertext encrypted by each cryptographic algorithm is $F$, and a set of test ciphertext files $CT_1, CT_2, \ldots, CT_s$ with unknown encryption algorithm.

Output: Classification results. Classification results of the test ciphertext files, they are algorithms in $at_1, at_2, \ldots, at_s$ ($at_1, at_2, \ldots, at_s \in A$).

The training stage includes the following 12 steps:

Step 1. Input a total of $k * F$ ciphertext files. According to the randomness test methods, we extract features from each ciphertext file, and obtain $k * F$ sets of ciphertext feature sets $FeaTr = \{feaTr_i^j | i = 1, 2, \ldots, k * F, j = 1, 2, \ldots, d\}$, where $feaTr_i^j$ represents the $j$-th feature of the $i$-th training ciphertext file.

Step 2. The feature set and ciphertext label form a sample set $(FeaTr, Lab)$ as the original data set, which is recorded as $T$.

Step 3. Input feature set $T$. There are $k * F$ ciphertext files in total, each ciphertext file represents a sample and each sample has $d$ features.

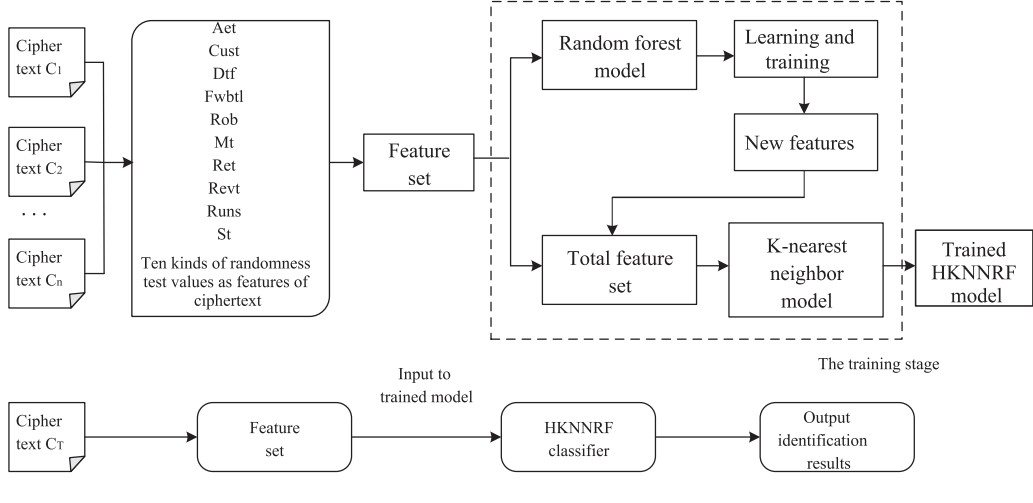

**Figure 3** **The flow of the cryptographic algorithm identification scheme constructed based on the HKNNRF model.**

Step 4. Use the method of sampling with replacement, extract $M$ samples from the feature set $T$ to form a sample set called Bootstrap sample set $T^*$, and it is used as root node samples of a decision tree.

Step 5. Among the $d$ features of the sample set, we select $t$ attributes randomly as candidate attributes, and calculate the best splitting attributes.

Step 6. Based on each value type of the best splitting attribute, segment the current data set to obtain the sub data sets.

Step 7. For each sub data set, selected $t$ attributes randomly, select the best splitting attribute, and segment the current data set at this level.

Step 8. Repeat steps 4–7 to establish $K$ decision trees to build a random forest and obtain a classification data set.

Step 9. Use the trained decision trees to construct new features and add them to the original features to get final features.

Step 10. Use one-hot encoder to normalize the final features.

Step 11. Use the normalized features to train KNN classifier.

Step 12. Output the integrated classifier HKNNRF.

The test stage includes the following two steps:

Step 1. Use NIST randomness test methods to scan the ciphertext files $CT_1$, $CT_2$, ..., $CT_s$ to be identified to obtain the ciphertext feature $FeaTr = \{feaTr_i^j | i = 1, 2, ..., s, j = 1, 2, ..., d\}$.

Step 2. Input the ciphertext feature $FeaTr = \{feaTr_i^j | i = 1, 2, ..., s, j = 1, 2, ..., d\}$, where $feaTr_i^j$ represents the $j$-$th$ feature of the $i$-$th$ test ciphertext file, into the trained HKNNRF and output the classification result of the ciphertext file, the result is cryptographic algorithm label in $at_1$, $at_2$, ..., $at_s$. The flow of the cryptographic algorithm identification scheme based on the HKNNRF model is shown in Fig. 3.

**Table 1 Specific parameter list of five block cipher algorithms.**

| Algorithm | Structure | Key | Mode | Parameter | Implementation |
|-----------|-----------|-------|------|-----------|----------------|
| AES | SP | Fixed | ECB | Fixed | Crypto |
| 3DES | Feistel | Fixed | ECB | Fixed | Crypto |
| Blowfish | Feistel | Fixed | ECB | Fixed | Crypto |
| CAST | Feistel | Fixed | ECB | Fixed | Crypto |
| RC2 | Feistel | Fixed | ECB | Fixed | Crypto |

# EXPERIMENTAL ENVIRONMENT

## Data preparation

The important features of ciphertext are the returned values of different NIST randomness test methods. In our work, we use the open source tool sp800_22_tests-master in python, which contains a variety of NIST randomness test methods, to extract the ciphertext features. Each of these different test methods is an independent module, works separately, and executes in parallel. For a single ciphertext file, the following is a complete feature extraction process: The main Python program read the ciphertext file, and inputs it into each randomness test method, then we can get different feature values of this ciphertext, save them in another corresponding file. Repeat the same operation on the remaining ciphertext files, and we can get the complete ciphertext feature files.

This article uses the random numbers generated by the Fortuna Accumulator method of Crypto encryption module in python as plaintexts, and then uses the cryptographic algorithm library in the same module to encrypt the plaintexts to obtain the ciphertexts. The encryption key and initialization vector are generated by the Cipher encryption module of Crypto (*Gharehchopogh & Abdollahzadeh, 2021*), and we use AES, 3DES, Blowfish, CAST and RC2 from cryptographic algorithm library to generate ciphertexts using a fixed 16-bit string key in ECB mode. We chose five ciphertext sizes: 1, 8, 64, 256, 512 KB, and each with 100 files, totaling 500 files for one cryptographic algorithm, so 2,500 files for five algorithms. We then used 15 different randomness test methods to extract features of ciphertexts; although we obtained 15 different values, only 10 useful values were selected as the final input of the classifier. Each ciphertext sample corresponds to a set of 10 features. Next, we divided the data set, selected 80% samples randomly as the training set, and took the remaining 20% as the test set. Moreover, there were $C_5^2 = 10$ kinds of binary-classification identification experiments. The specific parameters used by the five cryptographic algorithms are shown in Table 1.

## The evaluation criteria for classification results

In classification problems, the most commonly used criteria are:

(1) Accuracy: proportion of correctly classified samples to the total number of samples.

(2) Precision: proportion of the samples that are predicted to be positive by the model that are actually positive to the samples that are predicted to be positive.

**Table 2 Confusion matrix of classification results.**

| Real situation | Forecast result | |
| --- | --- | --- |
| | Positive | Negative |
| Positive | TP (True positive) | FN (False negative) |
| Negative | FP (False positive) | TN (True negative) |

(3) Recall: the proportion of samples that are actually positive that are predicted to be positive among the samples that are actually positive.

We used the confusion matrix to evaluate the classification results, and it produced four results, namely True Positive (TP), True Negative (TN), False Positive (FP) and False Negative (FN). Let TP, TN, FP, and FN respectively denote the corresponding sample numbers. The confusion matrix of the classification results is shown in Table 2.

The following shows how to calculate Accuracy, Precision, and Recall:

$$Accuracy = \frac{TP + TN}{TP + TN + FP + FN}.$$
$$Precision = \frac{TP}{TP + FP}.$$
$$Recall = \frac{TP}{TP + FN}.$$

In our article, we take accuracy as the main criteria.

## DICUSSION

To evaluate the performance of the proposed ensemble learning model on block cipher algorithm identification, we compared it with three classic machine learning models (SVM, KNN, RF). These models were applied to the same dataset, and the identification performance of binary-classification and five-classification were compared under different ciphertext file sizes (1, 8, 64, 256, 512 KB).

### Binary-classification identification on cryptographic algorithms

We establish a classification model based on the 10 kinds of useful features extracted in "data preparation" above. Then we calculated the accuracy, precision, and recall values of SVM, KNN, RF and HKNNRF models on different sizes of ciphertext file which were encrypted by AES and 3DES. The results are shown in Table 3.

The first column of Table 3 shows the evaluation indicators of the identification result, and the second column shows the sizes of ciphertext file. From Table 3, it can be concluded that the average identification accuracy of SVM, KNN, RF and HKNNRF on different sizes of ciphertext file are 0.565, 0.57, 0.595 and 0.695 respectively. In addition, the identification accuracy of HKNNRF model is affected by the size of ciphertext, When the ciphertext file size is 1 and 512 KB, the binary-classification identification accuracy on AES and 3DES is

**Table 3 Binary-classification experimental results based on ciphertext features.**

| Evaluating indicator | File size (KB) | Classifier | | | |
|---|---|---|---|---|---|
| | | SVM | KNN | RF | HKNNRF |
| Accuracy | 512 | 0.600 | 0.600 | 0.600 | 0.725 |
| | 256 | 0.575 | 0.575 | 0.625 | 0.650 |
| | 64 | 0.625 | 0.600 | 0.650 | 0.675 |
| | 8 | 0.525 | 0.525 | 0.575 | 0.700 |
| | 1 | 0.500 | 0.550 | 0.525 | 0.725 |
| Precision | 512 | 0.580 | 0.601 | 0.600 | 0.725 |
| | 256 | 0.580 | 0.583 | 0.628 | 0.700 |
| | 64 | 0.620 | 0.594 | 0.650 | 0.650 |
| | 8 | 0.530 | 0.532 | 0.600 | 0.675 |
| | 1 | 0.420 | 0.530 | 0.615 | 0.700 |
| Recall | 512 | 0.580 | 0.600 | 0.600 | 0.700 |
| | 256 | 0.580 | 0.575 | 0.625 | 0.650 |
| | 64 | 0.660 | 0.600 | 0.650 | 0.700 |
| | 8 | 0.520 | 0.525 | 0.575 | 0.650 |
| | 1 | 0.420 | 0.550 | 0.525 | 0.625 |

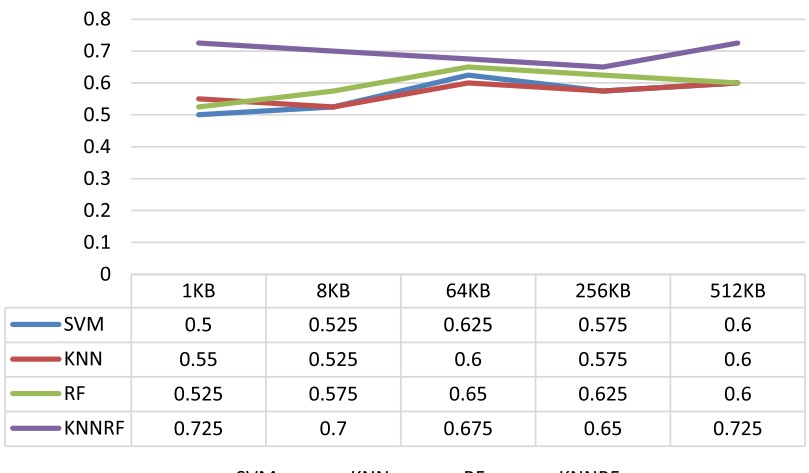

| | 1KB | 8KB | 64KB | 256KB | 512KB |
|---|---|---|---|---|---|
| SVM | 0.5 | 0.525 | 0.625 | 0.575 | 0.6 |
| KNN | 0.55 | 0.525 | 0.6 | 0.575 | 0.6 |
| RF | 0.525 | 0.575 | 0.65 | 0.625 | 0.6 |
| KNNRF | 0.725 | 0.7 | 0.675 | 0.65 | 0.725 |

**Figure 4 Identification accuracy under different file sizes.**

as high as 72.5%. Especially in the case of 1 KB, the identification accuracy of HKNNRF model is 22.5%, 17.5%, 20% higher than SVM, KNN and RF model respectively.

As shown in Fig. 4, it intuitively shows the identification accuracy of the binary-classification experiment using SVM, KNN, RF, HKNNRF model in the form of a line chart. Combining Table 3 and Fig. 4, it can be inferred that the cryptographic algorithm identification scheme based on ensemble learning has higher accuracy than the schemes based on single-layer machine learning model.

**Table 4 Binary-classification accuracy of five cryptographic algorithms based on HKNNRF.**

| Accuracy | 1 KB | 8 KB | 64 KB | 256 KB | 512 KB |
|---|---|---|---|---|---|
| 3DES and AES | 0.725 | 0.700 | 0.675 | 0.650 | 0.725 |
| 3DES and Blowfish | 0.625 | 0.625 | 0.650 | 0.625 | 0.650 |
| 3DES and CAST | 0.650 | 0.625 | 0.725 | 0.675 | 0.625 |
| 3DES and RC2 | 0.650 | 0.675 | 0.700 | 0.625 | 0.65 |
| AES and Blowfish | 0.600 | 0.700 | 0.675 | 0.650 | 0.675 |
| AES and CAST | 0.650 | 0.675 | 0.650 | 0.700 | 0.700 |
| AES and RC2 | 0.625 | 0.600 | 0.700 | 0.675 | 0.650 |
| Blowfish and CAST | 0.650 | 0.650 | 0.650 | 0.625 | 0.625 |
| Blowfish and RC2 | 0.700 | 0.600 | 0.675 | 0.700 | 0.700 |
| CAST and RC2 | 0.725 | 0.650 | 0.625 | 0.675 | 0.725 |

Next, continue to use HKNNRF to perform binary-classification identification experiments on the five algorithms of AES, 3DES, Blowfish, CAST and RC2, a total of 10 combinations. The identification results are shown in Table 4.

Figure 5 shows the contents of Table 4 in the form of histograms according to the difference of file size.

From Fig. 5, the binary-classification accuracy of the 10 combinations using HKNNRF model in five different file sizes can be more clearly observed, and we can find that all accuracy of HKNNRF are not less than 60%, and the highest can reach 72.5%. In addition, we can infer that the difference in the size of ciphertext file and the difference in the cryptographic algorithm used in ciphertext file will both affect the final accuracy.

## Five-classification identification on cryptographic algorithms

Finally, we established a five-classification model based on the 10 kinds of ciphertext features, and carry out identification experiment on AES, 3DES, Blowfish, CAST and RC2 using SVM, KNN, RF, HKNNRF in five different ciphertext file sizes. The results are shown in Table 5.

The experimental results show that the highest identification accuracy of HKNNRF can reach 34%, which is 13%, 12%, 11% higher than the accuracy of the single-layer KNN, single-layer RF and single-layer SVM model respectively. And the accuracy of HKNNRF are all not less than 24%. Similarly, as the size of ciphertext file changes, the accuracy changes accordingly, it also tells us that the size of ciphertext file has a certain affect on the accuracy.

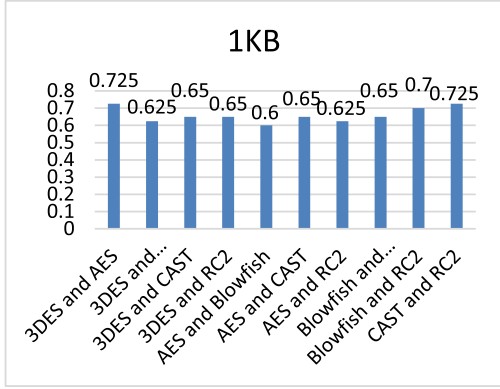

(a)    Binary-classification accuracy in 1KB case

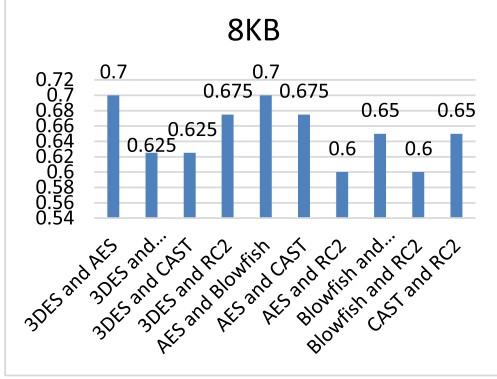

(b)    Binary-classification accuracy in 8KB case

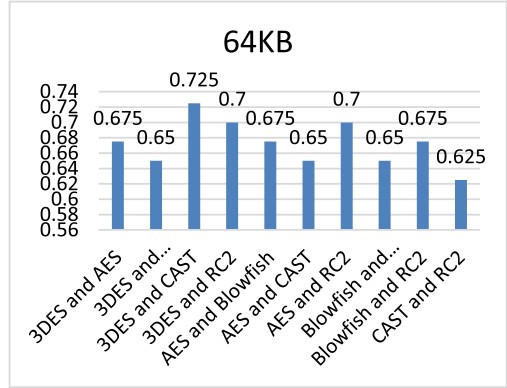

(c)    Binary-classification accuracy in 64KB case

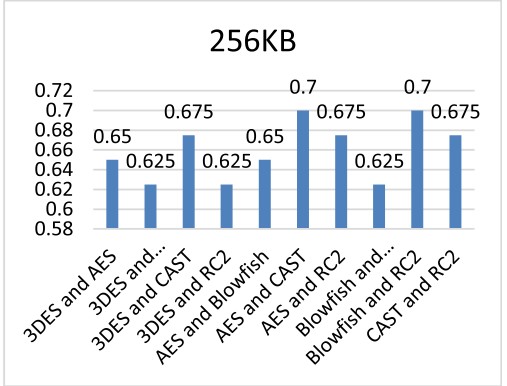

(d)    Binary-classification accuracy in 256KB case

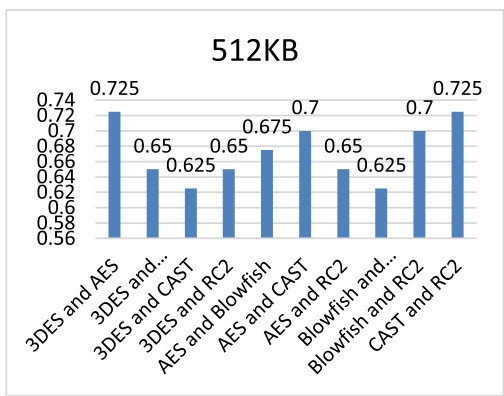

(e)    Binary-classification accuracy in 512KB case

**Figure 5  Comparison of binary-classification accuracy of different file sizes.**

**Table 5 Five-classification experimental results based on ciphertext features.**

| Evaluating indicator | File size (KB) | Classifier | | | |
|---|---|---|---|---|---|
| | | SVM | KNN | RF | HKNNRF |
| Accuracy | 512 | 0.190 | 0.200 | 0.170 | 0.240 |
| | 256 | 0.210 | 0.200 | 0.200 | 0.330 |
| | 64 | 0.100 | 0.160 | 0.220 | 0.270 |
| | 8 | 0.170 | 0.190 | 0.240 | 0.300 |
| | 1 | 0.230 | 0.210 | 0.220 | 0.340 |
| Precision | 512 | 0.230 | 0.219 | 0.210 | 0.298 |
| | 256 | 0.233 | 0.225 | 0.208 | 0.330 |
| | 64 | 0.115 | 0.099 | 0.231 | 0.227 |
| | 8 | 0.204 | 0.207 | 0.233 | 0.305 |
| | 1 | 0.214 | 0.222 | 0.223 | 0.371 |
| Recall | 512 | 0.190 | 0.200 | 0.170 | 0.240 |
| | 256 | 0.210 | 0.200 | 0.200 | 0.330 |
| | 64 | 0.100 | 0.160 | 0.220 | 0.270 |
| | 8 | 0.170 | 0.190 | 0.240 | 0.300 |
| | 1 | 0.230 | 0.210 | 0.220 | 0.340 |

## CONCLUSIONS AND FUTURE WORK

This article mainly focuses on the cryptographic algorithm identification problem. In the ciphertext-only scenario, we propose a new identification scheme based on ensemble learning named HKNNRF to achieve higher classification accuracy. The ciphertext files of different sizes encrypted by AES, 3DES, Blowfish, CAST and RC2 are used as identification objects to perform binary-classification and five-classification experiments, and the results show that the difference in the size of ciphertext file and the difference in the cryptographic algorithm used in ciphertext file both are factors that influence the identification accuracy. Take the binary-classification of AES and 3DES as an example; the average accuracy of HKNNRF can reach 69.5%, which is much higher than the average 56.5% accuracy of SVM, and also higher than the average 57%, 59.5% accuracy of KNN and RF model. In other binary-classification cases, the identification accuracy of HKNNRF are all not less than 60%, and the highest can reach 72.5%; In the five-classification case, the highest identification accuracy of HKNNRF can reach 34%, which is higher than the 21% accuracy of KNN, the 22% accuracy of RF and the 23% accuracy of SVM under the same experimental conditions, and the minimum identification accuracy of HKNNRF model is 24%. All results shown above mean the scheme based on ensemble learning has a higher accuracy compared with the scheme based on one single-layer classifier.

The model and scheme proposed in this article are mainly suitable for the identification of cryptographic algorithms. In rare cases, the identification performance of HKNNRF is not better than single-layer machine learning model. In the future, we plan to add classic asymmetric cryptographic algorithms such as RSA, ElGamal, and ECC as experimental objects to expand application scenarios. In addition, we will further study the identification

performance of HKNNRF model under different encryption modes of block cipher. As a new idea, the block cipher algorithm identification scheme based on ensemble learning is worthy of further exploration, and it has certain positive significance for the future research on block cipher algorithm identification.

### Funding

This work was supported by the National Natural Science Foundation of China (61972073, 61972215, 62066040); the Natural Science Foundation of Tianjin (20JCZDJC00640); the Key Specialized Research and Development Program of Henan Province (222102210062); the Basic Higher Educational Key Scientific Research Program of Henan Province (22A413004); and the National Innovation Training Program of University Student (202110475072). The funders had no role in study design, data collection and analysis, decision to publish, or preparation of the manuscript.

### Grant Disclosures

The following grant information was disclosed by the authors:
The National Natural Science Foundation of China: 61972073, 61972215, 62066040.
The Natural Science Foundation of Tianjin: 20JCZDJC00640.
The Key Specialized Research and Development Program of Henan Province: 222102210062.
The Basic Higher Educational Key Scientific Research Program of Henan Province: 22A413004.
The National Innovation Training Program of University Student: 202110475072.

### Competing Interests

The authors declare that they have no competing interests.

### Author Contributions

- Ke Yuan conceived and designed the experiments, analyzed the data, authored or reviewed drafts of the article, and approved the final draft.
- Daoming Yu performed the experiments, analyzed the data, performed the computation work, prepared figures and/or tables, authored or reviewed drafts of the article, and approved the final draft.
- Jingkai Feng performed the experiments, performed the computation work, prepared figures and/or tables, and approved the final draft.
- Longwei Yang conceived and designed the experiments, analyzed the data, authored or reviewed drafts of the article, and approved the final draft.
- Chunfu Jia analyzed the data, authored or reviewed drafts of the article, and approved the final draft.
- Yiwang Huang performed the computation work, prepared figures and/or tables, and approved the final draft.

## Data Availability

The raw data and code is available in the Supplemental Files.

## Supplemental Information

Supplemental information for this article can be found online at http://dx.doi.org/10.7717/peerj-cs.1110#supplemental-information.

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
