# Peer review of "A block cipher algorithm identification scheme based on hybrid k-nearest neighbor and random forest algorithm"

_PeerJ Computer Science, doi:10.7717/peerj-cs.1110_

## Round 0.1 · original submission · Major Revisions

We have received two reports on this paper. The reviewers have some concerns that need to be addressed before further processing. Please provide a detailed response letter. Thanks.

Reviewer 1 ·

Basic reporting

This manuscript proposes a new hybrid k-nearest neighbour and random forest (HKNNRF) model and constructs a block cipher algorithm identification scheme based on it. We use NIST randomness test methods to extract ciphertext features in the ciphertext-only scenario. The HKNNRF model is used to construct an identification classifier to carry out binary classification and multi-classification experiments on the block cipher algorithms. Moreover, they should emphasize that the multi-classification investigation is to classify five different block cipher algorithms. Experiments show that when the ciphertext size and other experimental conditions are the same, compared with the scheme based on the single-layer classifier model, the HKNNRF model has a higher classification accuracy and stability. Adequate revisions to the following points should be undertaken to justify the recommendation for publication.
• This paper has more than spelling and grammatical errors. Please fix all of them. And some of the sentences are not understood.
• The abstract section is long. Please rewrite an abstract section, justify an obtained result and contribution, improve a proposed method, etc.
• Please write your contribution to this paper in the Introduction section.
• The authors should clearly state the limitations of the proposed method in other applications.
• Please draw a flowchart of the proposed method.
• Please use a new algorithm for comparisons, such as the Farmland fertility algorithm, African Vultures Optimization Algorithm, and Artificial Gorilla Troops Optimizer.
• Please write the paper structure in the end paragraph of the Introduction section.
• Please change the title of the end section (Conclusion) to (Conclusion and Future Works) and write some future work on your proposed method.
• All figures have low quality, and please improve all of them.
• Related work is missed; the authors used some new and SCI or Scopus indexed papers in this section.
Good luck

Experimental design

...

Validity of the findings

...

Additional comments

...

Reviewer 2 ·

Basic reporting

(i)There are some places in the paper, where the English is poor and hence the reader is unable to comprehend what the authors intend to say.
eg: line numbers-264, 265, 298(This thesis uses),. Also the line "Moreover, what we should emphasize is that the multi-classification experiment is to classify five different block cipher algorithms in our paper." should be re-written as it lacks clarity. The abstract to be re-written clearly in Professional English.
(ii) Sufficient Literature and back ground work has been provided.
(iii) Adequate raw data shared and the paper structure is good.

Experimental design

Adequate information is given paper about the investigation performed.

Validity of the findings

All the data and code has been provided. However in the Discussion section, the comparison of the proposed method with an existing method from literature has to be done to understand how this work is better than previous works.

---

## Round 0.2 · accepted · Accept

The reviewers' comments have been addressed. The paper can be accepted. Congratulations.

Reviewer 1 ·

Basic reporting

From the response letter, I think the paper has been well revised according to the previous reviewers, and the current version of the manuscript is acceptable for publication.

Experimental design

From the response letter, I think the paper has been well revised according to the previous reviewers, and the current version of the manuscript is acceptable for publication.

Validity of the findings

From the response letter, I think the paper has been well revised according to the previous reviewers, and the current version of the manuscript is acceptable for publication.

Additional comments

From the response letter, I think the paper has been well revised according to the previous reviewers, and the current version of the manuscript is acceptable for publication.

Reviewer 2 ·

Basic reporting

no comments

Experimental design

no comment

Validity of the findings

no comment

Additional comments

no comment